# Conservative & Aggressive NaNs Accelerate U-Nets for Neuroimaging

## Abstract

Advancements in deep learning for neuroimaging have resulted in the development of increasingly complex models designed for a wide range of tasks. Despite significant improvements in hardware, enhancing inference and training times for these models remains crucial. Through an analysis of numerical uncertainty in convolutional neural networks (CNNs) inference, we found that a substantial amount of operations in these models are applied to values dominated by numerical noise, with little to no impact on the final output. As a result, up to two-thirds of the floating-point operations executed by some CNNs appear unnecessary. To address this inefficiency, we introduce Conservative & Aggressive NaNs —novel variations of PyTorch's max pooling and unpooling operations. These techniques identify numerically unstable voxels and replace them with NaNs, allowing models to bypass operations on irrelevant data. We evaluated Conservative & Aggressive NaNs on four models: the FastSurfer and FONDUE CNNs, widely used neuroimaging tools, the Xception CNN, an image classification model, and another CNN designed to classify the MNIST dataset. We observed speedups for data containing at least 50% NaNs, and most notably, for data with more than two-thirds NaNs (as in many of our use cases), we observed an average speedup of $1.67\times$. Conservative NaNs reduces the number of convolutions by an average of 30% across all tested models and datasets, with no measurable degradation in performance. In some model layers, it can skip up to 64.64% of convolutions with no performance degradation. The more proactive Aggressive NaNs approach can skip up to 69.30% convolutions for FastSurfer with no performance degradation, however, it sometimes leads to measurable performance degradation for FONDUE and MNIST. Overall, Conservative & Aggressive NaNs provide substantial opportunities for runtime acceleration of inference in CNNs, which could potentially reduce the environmental impact of these models.

## 1 Introduction

Convolutional Neural Networks (CNNs), in particular U-Nets, are transforming neuroimaging by progressively replacing traditional image analysis software with models that deliver comparable performance in a fraction of the runtime. This allows for scalable processing of large datasets. However, optimizing the inference and training times of these models remains a critical challenge, as improvements in this area could facilitate near-real-time analyses across various applications, support the training of larger models for tasks previously considered computationally infeasible, and reduce the environmental impact of such analyses.

While investigating the numerical uncertainty in CNNs, we identified a key inefficiency in the pooling and unpooling operations: approximately two-thirds of the embedding values propagate pure numerical noise, leading to unnecessary computations in subsequent convolutional layers. This issue arises when max pooling is applied to windows containing multiple near-equal values (within a small epsilon threshold), leading to instability in the selection of the maximum index. When unpooling later restores values based on these indices, small inconsistencies in index selection can result in numerical noise filling large portions of the embedding space (Figure 1).

Despite the prevalence of this instability, affected models continue to train successfully and produce accurate predictions. This suggests that much of the processed noise is irrelevant to model perfor-

mance, presenting an opportunity for computational efficiency. To leverage this insight, we introduce the Convervative & Aggressive NaNs approaches, which modify pooling operations as follows:

- Conservative NaNs modifies max pooling to return all possible indices of the maximum value. When the unpooling operation is called, NaNs are inserted at these indices.

- Aggressive NaNs takes a more aggressive approach, preventing unstable indices altogether by modifying max pooling to output NaNs in case of numerical uncertainty.

In both methods, we adapt convolution operations to handle tensors with NaNs, skipping computations when NaNs exceed a predefined threshold.

## 2 NUMERICAL ANALYSIS OF FASTSURFER SEGMENTATION MODEL

### 2.1 NUMERICAL UNCERTAINTY OF FASTSURFER EMBEDDINGS

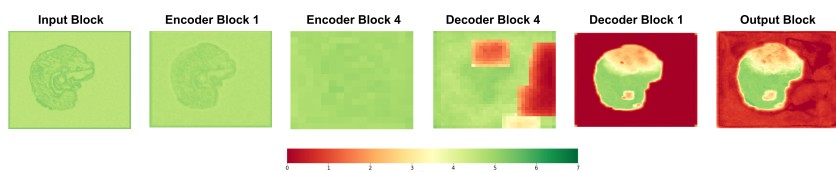

Figure 1: Significant Digit Maps for FastSurfer Model Embeddings in Selected Model Layers for Numerically Unstable Data.

We examined the numerical uncertainty of the FastSurfer CNN during inference, focusing on the stability of the final classification results as well as the embeddings.

To do so, we used Monte Carlo Arithmetic (MCA) Parker (1997)—a stochastic arithmetic technique that introduces random numerical perturbations to assess numerical uncertainty—implemented through the Verrou tool Févotte & Lathuiliere (2016) that dynamically instruments binary executables with MCA. Using Verrou, we instrumented FastSurfer inference to generate 10 iterations of the model's embeddings, each subjected to random perturbations, and computed the number of significant digits across the 10 iterations Sohier et al. (2021). This analysis was performed for every layer of the FastSurfer model, with the results visualized as heatmaps in Figure 1. This revealed that a large fraction of the model embeddings were purely numerical noise (zero significant digits), represented by red-colored regions in the figure.

The instability first appeared during the max unpooling operations in the FastSurfer decoder, which we later determined resulted from the indices provided to the max unpooling operation. This instability becomes especially pronounced when upsampling is applied to regions of the image background, where uniform values dominate.

Interestingly, the segmentations resulting from the model were still accurate in spite of the presence of substantial numerical noise in the embeddings, which suggested that computations performed on these values were not contributing to the final result. This observation motivated the design of NaN Convolutions and Conservative & Aggressive NaNs presented in this paper.

### 2.2 NUMERICAL UNCERTAINTY IN MAX POOLING

Max pooling Boureau et al. (2010) is a widely used downsampling technique that replaces a defined window of values with its maximum value. It can optionally return indices that indicate the original locations of these maximum values. During upsampling, max unpooling uses these indices to restore the maximum values to their original positions, filling the remaining voxels with zeros. This process ensures that the spatial structure of the input data is partially reconstructed based on the locations of the selected maximum values. The indices generated during max pooling are especially useful in U-Net architectures, where downsampling and upsampling processes are frequently coupled Zeiler et al. (2010); Çiçek et al. (2016); Lu et al. (2019); Plascencia et al. (2023); De Feo et al. (2021).

While investigating the numerical uncertainty of CNNs during inference, we found that numerical instabilities arose during max unpooling operations due to fluctuations in the indices used by unpooling. When values within a pooling window are close to each other, even slight noise can lead to index shifts while the maximum value remains unchanged. This instability is particularly evident when upsampling is applied to areas of an image's background, where values are nearly uniform. Interestingly, we observed that the propagation of this numerical noise did not adversely impact the final outputs of the models we tested.

Unstable voxels contribute no meaningful information to the model. To address this inefficiency, we propose Conservative & Aggressive NaNs as a way to bypass operations on such irrelevant voxels. In floating-point arithmetic, NaNs (Not-a-Number) are special values defined by the IEEE 754 standard to represent undefined or unrepresentable results. A NaN is represented by an exponent of all ones and a non-zero mantissa, and it is used to represent undefined or exceptional results in floating-point calculations. Leveraging this concept, we use NaNs to mark numerically irrelevant voxels, effectively skipping over operations that would otherwise be wasted on data that provides no useful information. This approach enhances computational efficiency by allowing the model to focus on relevant data, without altering the final output.

# 3 CONSERVATIVE NANS

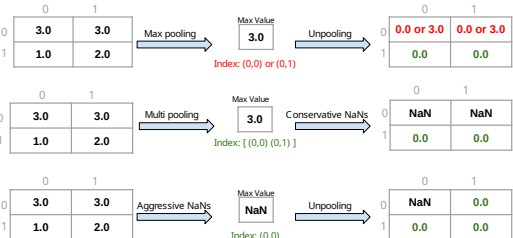

Figure 2: Comparison of Max Pooling vs Conservative & Aggressive NaNs in the presence of numerical uncertainty. Green indicates numerically stable values, red unstable values.

Conservative NaNs is illustrated in Figure 2. Given $\mathbf{X}$, the input tensor, $\mathbf{W}$, a sliding window defined by kernel size $k \times k$ and stride $s$, we first perform Multi Pooling to extract all indices of the max values per window. Given $\mathbf{W}$, $\max(\mathbf{W})$ denotes the maximum value in $\mathbf{W}$ and $\mathrm{idx}(\mathbf{W})$ are the indices of $\mathbf{W}$. We define the set of indices for all maximum values:

$$S = \{i \in \mathrm{idx}(\mathbf{W}) \mid |W_i - \max(\mathbf{W})| < \epsilon\}$$

This ensures that, rather than returning a single maximum index, the function returns a set $S$ containing all positions where the values are equal to the maximum value within an $\epsilon$ tolerance of $10^{-7}$. This threshold corresponds to the resolution of single-precision computations and is applied per $\mathbf{W}$. This gives us $\mathbf{M} = \{m_1, m_2, \ldots, m_n\}$, a tensor of max values from $\mathbf{X}$ which is identical to normal max pooling and $\mathcal{D}$, a dictionary mapping each max value $m_i$ to a set of indices $S$ in $\mathbf{X}$.

We then apply the Conservative NaNs operation:

$$\mathbf{Y}[j] = \begin{cases} m_i & \text{if } j \in S \text{ and } |S| = 1 \\ \mathrm{NaN} & \text{if } j \in S \text{ and } |S| > 1 \\ 0 & \text{otherwise} \end{cases}$$

where $\mathbf{Y}$ is the upsampled output tensor containing the max values in their original location prior to pooling as well as introducing NaNs.

# 4 AGGRESSIVE NANS

We illustrate Aggressive NaNs in Figure 2. For each tensor window $\mathbf{W}$ in the input tensor $\mathbf{X}$, the max pooling operation $Y(\mathbf{W})$ is computed as:

$$Y(\mathbf{W}) = (m, i_m)$$

Where $m$ is the maximum value of $\mathbf{W}$ and $i_m$, the index of the maximum value of $\mathbf{W}$, i.e. $i_m = \text{argmax}(\mathbf{W})$.

For Aggressive NaNs, we redefine the max pooling operation $Y$ to handle potential NaNs and tie-breaking for repeated maximum values as follows:

$$Y'(\mathbf{W}) = \begin{cases} (\text{NaN}, (0,0)) & \text{if Counter} > t_1 \\ (m, i_m) & \text{otherwise} \end{cases}$$

Where $\text{Counter} = \text{Card}(\{w \in \mathbf{W}_{:,:,j} \mid |w - m| < \epsilon\})$, $t_1$ is a user-defined threshold that specifies the maximum number of near-equal values allowed for $m$, and $\epsilon = 10^{-7}$ is the tolerance to handle floating-point precision issues. We assign index $(0,0)$ for unstable pooling cases to simplify implementation. This default is efficient and robust to implement, avoiding the overhead of resolving tie-breaking logic. We derive $m$ and $i_m$, from $\bar{\mathbf{W}}$, defined as $\bar{\mathbf{W}} = \{W_{n,i,j} \in \mathbf{W} \mid W_{n,i,j} \text{ is not } NaN\}$ in order to ignore NaN values.

## 5 NaN Convolution

NaN Convolution handles the presence of NaNs introduced through Aggressive or Conservative NaNs, skipping over numerically irrelevant operations. Consider a padded 4D input tensor $\mathbf{X}$ of shape $(N, C_{in}, H_{in}, W_{in})$, a 4D kernel tensor $\mathbf{K}$ of shape $(C_{out}, C_{in}, H_k, W_k)$, and a NaN threshold $t_2 \in [0, 1]$, where $N$ is the batch size, $C_{in}$ is the number of input channels, $C_{out}$ is the number of output channels, $H_{in}$ is the height of the input, $W_{in}$ is the width of the input, $H_k$ is the height of the kernel, and $W_k$ is the width of the kernel.

For each window $\mathbf{W}$ in the input tensor, where $\mathbf{W}$ is of shape $(C_{in}, H_{in}, W_{in})$ and its elements are in $\mathbb{R} \cup \{\text{NaN}\}$, we define the output of the NaN convolution of $\mathbf{W}$ by kernel $\mathbf{K}$ as performed per batch:

$$Y_{c,h,w} = \begin{cases} \text{NaN} & \text{if } r_{c,h,w} \geq t_2 \\ \sum_{c=0}^{C_{in}-1} \sum_{h=0}^{H_k-1} \sum_{w=0}^{W_k-1} \bar{W}_{c,h,w} \, K_{c,h,w} & \text{if } r_{c,h,w} < t_2 \end{cases}$$

Where $r_{c,h,w}$ is the ratio of NaNs across the input channels, height and width dimensions:

$$r_{c,h,w} = \frac{\text{Card}(\{w \in \mathbf{W}_{n,i,j} \mid w = \text{NaN}\})}{C_{in} H_{in} W_{in}}$$

We apply this NaN Convolution to all convolution layers in the network. An exception is made for pointwise convolutions, which are addressed with a modified approach detailed in Appendix A.

We define $\bar{\mathbf{W}}$ as the modified window where NaNs are replaced with one of two approaches.

**Approach A**   replaces NaNs with $\mu_{n,i,j}$, defined as the mean of the non-NaN values within $\mathbf{W}$:

$$\bar{\mathbf{W}} = \begin{cases} \mu_{n,i,j} & \text{if } \mathbf{W}_{n,i,j} = \text{NaN} \\ \mathbf{W}_{n,i,j} & \text{otherwise} \end{cases}$$

**Approach B**   replaces NaNs with a random value from a Gaussian distribution centered around $\max_{n,i,j}$, the maximum of the non-NaN values within $\mathbf{W}$, and a standard deviation $\sigma$ of $10^{-3}$.

$$\bar{\mathbf{W}} = \begin{cases} x \sim \mathcal{N}\left(\max_{n,i,j}(\mathbf{W}), \sigma^2\right) & \text{if } \mathbf{W}_{n,i,j} = \text{NaN} \\ \mathbf{W}_{n,i,j} & \text{otherwise} \end{cases}$$

Approach A can smooth the outputs of NaN Convolutions, which is advantageous in some settings. However, when smoothing is undesirable, Approach B introduces variability into the output. This variability is particularly useful in models with subsequent iterations of Aggressive NaNs, as it prevents overly aggressive NaN introduction that could result from exploiting the smoothed output of Approach A.

$\bar{\mathbf{W}}$ is introduced to ensure that regions where the number of NaNs remains below the threshold $t_2$ are unaffected, since standard operations cannot inherently manage NaN values. It replaces the previous versions of the window and serves as the basis for the convolution operation.

## 6 EXPERIMENTS & RESULTS

We implemented Conservative & Aggressive NaNs, NaN Convolutions and standard convolutions in Python and evaluated their performance across four convolutional neural networks: FastSurfer Henschel et al. (2020), a U-Net for whole-brain segmentation; FONDUE Adame-Gonzalez et al. (2023), a nested U-Net for MRI denoising; Xception Chollet (2017), an image classification model based on depthwise separable convolutions; and a classic CNN for digit classification on MNIST LeCun & Cortes (1998). We provide architectural diagrams for all models in Appendix C and dataset description in Appendix B. Although we primarily report results on the axial brain plane, we verified that similar conclusions hold across the sagittal and coronal planes.

To quantify the computational impact, we calculated the ratio of skipped operations relative to the total number of convolutions. This ratio was tracked both across individual operations, the architectural layers of the models and across brain slices in the neuroimaging data, providing a comprehensive view of the effect of the NaN approaches. We experimented with a range of threshold values for $t_2$ (from 1.0 to 0.4), which determine the minimum ratio of NaNs in a patch required to skip the corresponding convolution. A threshold of 1.0 skips only fully-NaN patches, while 0.4 allows for more aggressive skipping. Unless otherwise specified, we refer to the baseline models that use standard PyTorch operations as the "default" implementations. Scripts, configuration details, and further documentation for this experiment are available on GitHub nan.

### 6.1 NaN CONVOLUTIONS ACHIEVE SPEEDUP

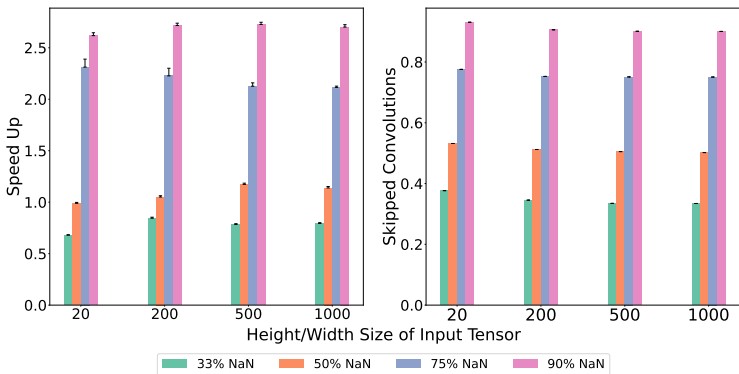

Figure 3: Average Speed Up Between Standard and NaN Convolutions (left plot). Ratio of skipped convolutions (right plot).

We measured runtime performance across a range of input tensor sizes, where NaNs were randomly distributed. The NaN Convolution threshold $t_2$ was fixed at 0.5 throughout to isolate the effects of varying tensor size and NaN density. This threshold was selected based on its consistent success in multiple real-world neuroimaging use cases. We define speedup as the ratio of standard convolution time to NaN Convolution time; values above 1.0 indicate a performance gain. Nonetheless, depending on the size of the input and NaN density, we had compute times ranging from $3.31 \times 10^{-3}$ to 9.16 seconds per NaN Convolution and a range of $8.67 \times 10^{-3}$ to 7.29 seconds per standard convolution.

As shown in the left plot of Figure 3, under random NaN distributions, speedups were modest at 33% NaN presence ($0.68\times$–$0.84\times$), but surpassed parity by 50% ($1.05\times$ average). At 75% and 90% NaN presence, speedups consistently exceeded $2\times$, reaching up to $2.76\times$ for large matrices. These results confirm that skipping NaN regions yields significant runtime improvements as input sparsity increases. However, a plateau was observed at larger inputs, likely due to shared bottlenecks in resource bandwidth, or the Python interpreter, which affect both convolution types equally.

Furthermore, we translate the NaN density in input tensors into the number of skipped convolutions in the right plot of Figure 3, and observe that the NaN density is generally proportional to the number of skipped convolutions. Crucially, input tensors containing 50% or more NaNs are common in real-world neuroimaging pipelines such as FastSurfer and FONDUE. In such cases, NaN Convolution offers a practical strategy to bypass unnecessary computation without compromising model output.

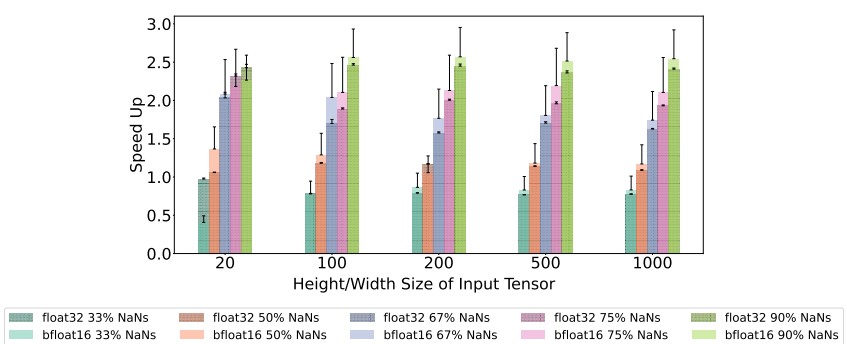

Figure 4: Comparison of Standard and NaN Convolutions in bfloat16 and float32 for GPU

To further assess the generalizability of our method, we evaluated its compatibility with reduced-precision settings. As shown in Figure 4, NaN Convolution maintained its performance advantage over standard convolution when executed in lower-precision formats such as bfloat16 on GPU. In fact, we observe a slight increase in speed up for bfloat16 compared to float32 on GPU, which we attribute to hardware-level optimization for bfloat16. Compared to quantization and pruning, NaN-based operations are orthogonal and complementary, providing an additional pathway to runtime efficiency in high-resolution, sparse-input settings. Exploring combined strategies remains a compelling direction for future work. Overall, these findings show that even without hardware-level acceleration, NaN Convolution offers a substantial and practical runtime advantage when large portions of the input are impacted by numerical noise.

## 6.2 Conservative NaNs Skips 30% of Convolutions in Neuroimaging Models and Preserves Accuracy

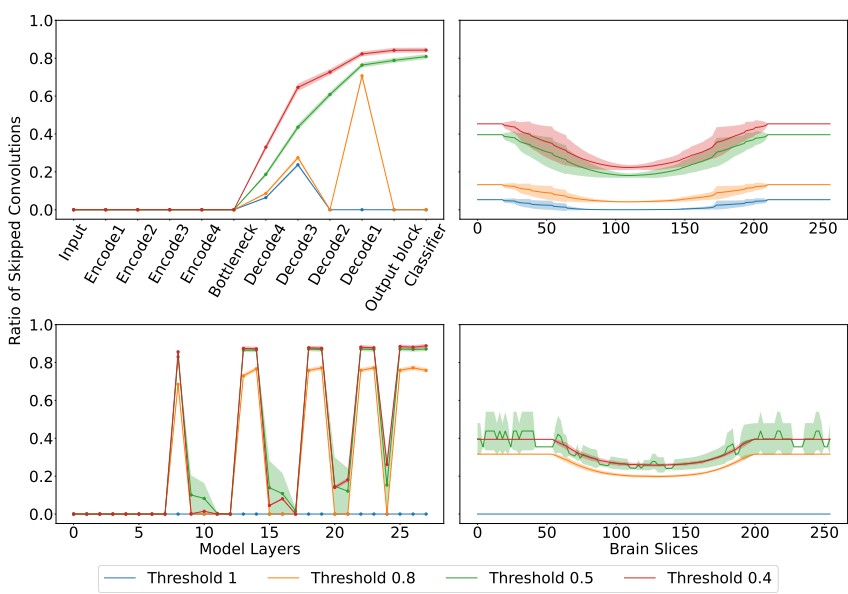

Figure 5: Ratio of Skipped Convolutions Across FONDUE (top) and FastSurfer (bottom) for Architecture (left) and Axial Brain Slices (right). For Thresholds 1, 0.8, 0.5 and 0.4, 6.1 %, 10.95%, 26.59% and 31.95% of total convolutions were skipped respectively for FastSurfer. For FONDUE, 0 %, 26.85%, 33.97% and 33.77 % of total convolutions were skipped respectively.

Conservative NaNs targets voxels in the numerically unstable background of neuroimaging data, which can account for up to two-thirds of the input space. By skipping convolutions on these voxels, the method reduces redundant computation without harming downstream predictions. Importantly,

once the proportion of skipped convolutions surpasses 50%, we estimate a theoretical speedup of up to 2×, calculated by translating the skip ratios into runtime savings as shown in Figure 3.

In FastSurfer, skipped operations occur exclusively in decoder blocks, with the fraction increasing from mid to late layers (Figure 5, top left). Slice-wise analysis shows the highest skip rates at the extremes of the brain volume, where background voxels dominate (Figure 5, top right). Across thresholds of 1, 0.8, 0.5, and 0.4, total computational savings were 6.12%, 10.95%, 26.59%, and 31.94%, respectively, rising to 16.21–64.64% when restricted to decoder layers.

In FONDUE, a nested U-Net architecture model, skipped convolutions also concentrated in background regions (Figure 5, bottom row). At threshold 0.5, variability increased because NaNs appeared inconsistently across slices. Unlike FastSurfer, where NaNs propagated into the final output (but could be safely replaced with zeros since they only affected background voxels), FONDUE naturally limited NaN propagation due to fewer unpooling operations. Its hierarchical structure caused NaNs to emerge progressively from deeper to shallower layers rather than monotonically, resulting in overall skip ratios of 0%, 26.85%, 33.97%, and 33.77% for thresholds of 1, 0.8, 0.5, and 0.4.

We then assessed performance with each model's standard evaluation metrics. For FastSurfer, we analyzed the Dice-Sørensen coefficient Dice (1945) to quantify the spatial overlap between FastSurfer's segmentations and the reference segmentations produced by the FreeSurfer pipeline Fischl et al. (2002). Since FastSurfer was trained on data processed by FreeSurfer, these reference segmentations serve as the ground truth for assessing its performance. In Figure 6, we observe the Dice–Sørensen scores were unchanged across thresholds; accuracy matched the default implementation, with visible declines near the brain center attributable to anatomical complexity, not Conservative NaNs.

For FONDUE, we evaluated peak signal-to-noise ratio (PSNR), which quantifies image quality, and corroborated our findings with the structural similarity index (SSIM) metric, which captures perceptual differences in luminance, contrast, and structure (not reported here to avoid redundancy). In the left plot of Figure 7, PSNR was maintained across thresholds 1–0.4. Thresholds <0.4 degraded performance and were excluded. Overall, Conservative NaNs preserves model performance while significantly reducing computational workload.

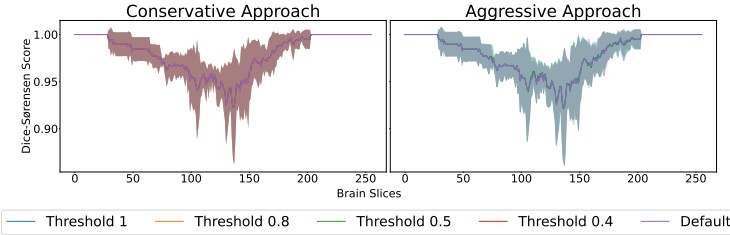

Figure 6: Comparison of Dice-Sørensen scores for default, Conservative (left plot) & Aggressive NaNs (right plot) Implementations of Fastsurfer across Axial Brain Slices.

### 6.3 Aggressive NaNs Trades Off Accuracy with Increased Efficiency

Aggressive NaNs skips a significant fraction of convolution operations, but this efficiency gain comes with a trade-off in accuracy. We evaluated the method on four different models—FastSurfer, FONDUE, MNIST, and Xception—covering both neuroimaging and natural image tasks. Since

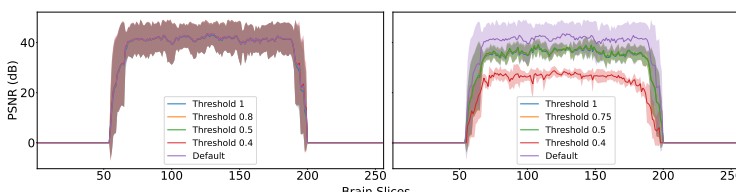

Figure 7: Comparison of PSNR for Default and Conservative NaNs (left plot) and Default and Aggressive NaNs (right plot) Across Axial Brain Slices for FONDUE

unpooling layers are absent in Xception and MNIST, Conservative NaNs was applied only to FastSurfer and FONDUE, but Aggressive NaNs were applied to all models. For NaN substitution during convolution, we used Approach A for the neuroimaging and Xception CNNs, and Approach B for the MNIST CNN, as this setup yielded the most robust performance. Overall, we find that Aggressive NaNs is best suited for models operating on homogeneous image regions, such as those found in MRI, while its benefits diminish on heterogeneous RGB datasets.

FastSurfer consistently maintained strong performance despite extensive skipping of convolutions. 33.24% of total convolutions were skipped at threshold 1, and 44.19% at threshold 0.5. When focusing only on NaN-affected layers, the rate of skipped operations rose to 50.59% and 69.30%, respectively—demonstrating the aggressive efficiency gains possible with moderate thresholds. As shown in Figure 8, up to 44.19% of total convolutions were skipped at threshold 0.5, with even higher rates in NaN-affected layers—corresponding to a theoretical speedup approaching $2\times$. Dice scores remained nearly identical to the default model, with only minor deviations in the cerebellum region. Since FastSurfer is trained on FreeSurfer outputs, and FreeSurfer is known to segment the cerebellum inaccurately Morell-Ortega et al. (2024); Carass et al. (2018); Romero et al. (2017), the observed drop is likely due to unreliable labels rather than Aggressive NaNs itself (see Appendix D for a detailed analysis). This result demonstrates that Aggressive NaNs can deliver substantial efficiency improvements without sacrificing segmentation quality in robust neuroimaging architectures.

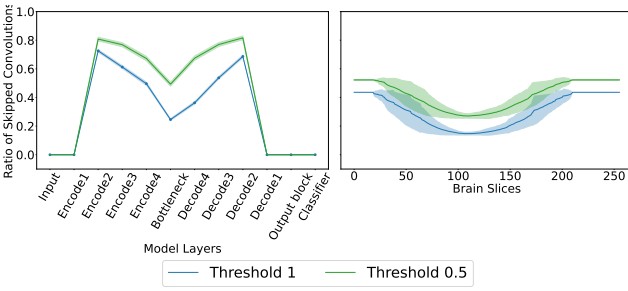

Figure 8: Ratio of Skipped Convolutions Across FastSurfer for Architecture (left plot) and Axial Brain Slices (right plot). For Threshold 1 and 0.5, 33.24 % and 44.19 % of convolutions were skipped.

In contrast, FONDUE exhibited a measurable drop in accuracy (Figure 7). PSNR declined across thresholds, although most PSNR values remained above 20 dB, which is considered acceptable for MRI quality. This indicates that while Aggressive NaNs reduces computation, it can be model-dependent.

In the MNIST digit classification benchmark, Aggressive NaNs showcased its sensitivity to data redundancy. With high thresholds ($\geq 1$), accuracy remained near 99%, while lower thresholds highlighted the method's ability to aggressively prune computations—skipping up to 78.6% of convolutions, equivalent to a $\sim 2\times$ speedup in some layers (Figure 9a). However, this came with steep accuracy trade-offs at low thresholds, reflecting the limited redundancy of MNIST compared to larger neuroimaging datasets.

When applied to the Xception CNN on ImageNet, Aggressive NaNs further highlighted its dependence on data structure (Figure 9b). The method is demonstrated as compatible with depthwise separable convolutions while preserving accuracy, but only $\sim 1\%$ of convolutions were skipped, underscoring the reduced opportunity for efficiency gains in heterogeneous RGB images. These results position Aggressive NaNs as a strong candidate for domains with high spatial redundancy, while remaining technically applicable to more complex architectures.

Taken together, we show that Aggressive NaNs provides substantial computational savings on models trained for neuroimaging (e.g., FastSurfer) and, to a lesser extent, on simpler datasets like MNIST. While it sometimes trades efficiency for accuracy—degrading performance in sensitive architectures (FONDUE) and showing limited benefit on heterogeneous RGB datasets (MNIST, Xception)—we also show it works well beyond standard convolution implementations. Its strengths lie in domains with homogeneous image regions, provided thresholds and architectures are chosen carefully.

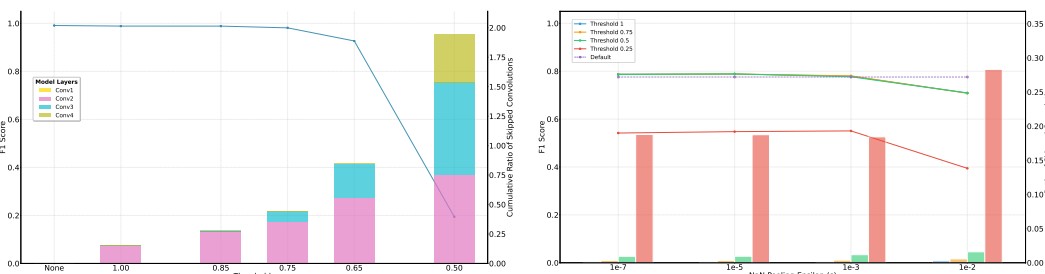

(a) F1 and cumulative ratio of skipped convolutions across thresholds for MNIST.

(b) F1 and ratio of skipped convolutions across $\epsilon$ values for $t_2$ thresholds for Xception CNN on ImageNet.

Figure 9: F1 (line plot) and convolution skipping patterns (bar plot) under Aggressive NaNs for (a) MNIST CNN and (b) Xception CNN.

## 7 CONCLUSION

We introduced Conservative & Aggressive NaNs, and NaN Convolutions—modified operations that skip computations on numerically unstable and irrelevant voxels by propagating NaNs. Conservative NaNs reacts to instability as it arises, while Aggressive NaNs proactively filters out irrelevant data, allowing selective computation based on input stability in CNNs and computational efficiency. Furthermore, NaN persistence can offer a mechanism for quantifying uncertainty, especially in ambiguous regions of medical images. This dual use—efficiency and interpretability—makes the proposed methods particularly valuable for clinical and explainable AI applications.

Our experiments on neuroimaging models (FastSurfer and FONDUE), classification benchmarks (MNIST CNN), and a depthwise separable CNN (Xception) reveal distinct behaviors for the two strategies. Conservative NaNs consistently preserved performance across all neuroimaging data, reducing computations by 31.94% in FastSurfer and up to 33.97% in FONDUE without any loss of accuracy. Aggressive NaNs, by contrast, is effective primarily on MRI data, achieving up to 39% convolution reduction in FastSurfer, but requires careful threshold tuning to avoid performance degradation. MNIST and Xception serve as clear examples of the limitations of these methods: while some efficiency gains are possible, heterogeneous RGB data and complex natural images limit the extent of skipped operations and the achievable speedup, despite stable NaN propagation in Xception's depthwise separable convolutions.

The NaN Convolution time trials further confirmed substantial runtime improvements—especially at high NaN densities—with consistent trends across float32 and bfloat16 precisions on CPU and GPU. These results establish a novel and practical path toward improving the computational efficiency of convolutional networks. Beyond theoretical contribution, we provide evidence of speedup and reduced resource utilization, which can be linked to lower resource consumption. In an era where the environmental cost of deep learning is under increasing scrutiny, our findings highlight how architectural-level innovations can contribute to more sustainable AI. Future directions include combining NaN-aware methods with sparse tensor representations, im2col-based transformations, and hardware-aware execution strategies to further increase efficiency. Together, these developments can pave the way for scalable, efficient, and environmentally conscious deep learning.

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

# A    POINTWISE CONVOLUTION EDGE CASE

Pointwise convolutions are typically the second step in depthwise separable convolutions. Also referred to as $1 \times 1$ convolution, it is used to combine the output channels across different feature maps.

For pointwise convolutions, our method functions as long as the number of input channels ($C_{in}$) is strictly greater than 1, ensuring that enough values exist to compute a meaningful ratio and propagate NaNs accordingly. Otherwise, if all three dimensions of the convolution window are 1, e.g. in a $1 \times 1$ convolution where $C_{in} = 1$, the kernel operates on a single scalar value. In this case, our default threshold-based NaN Convolution approach was not well defined. To address this, we've revised its definition as follows:

$$Y_{c,h,w} = \begin{cases} \text{NaN} & \text{if } \text{numel}(\bar{W}_{c,h,w}) = 1 \ \& \ \bar{W}_{c,h,w} = \text{NaN} \\ \sum_{c=0}^{C_{in}\text{-}1} \sum_{h=0}^{H_k\text{-}1} \sum_{w=0}^{W_k\text{-}1} \bar{W}_{c,h,w} \ K_{c,h,w} & \text{otherwise} \end{cases}$$

This extension ensures consistency with the intended behavior of NaN handling.

# B    DATASETS

Table 1: Subjects sampled in the CoRR dataset.

| Subject | Image Dimension | Voxel Resolution | Data Type |
|---------|-----------------|------------------|-----------|
| sub-0025248 | (208, 256, 176) | (1.00, 1.00, 1.00) | float32 |
| sub-0025531 | (160, 240, 256) | (1.20, 0.94, 0.94) | float32 |
| sub-0025011 | (128, 256, 256) | (1.33, 1.00, 1.00) | float32 |
| sub-0003002 | (176, 256, 256) | (1.00, 1.00, 1.00) | int16 |
| sub-0025350 | (256, 256, 220) | (0.94, 0.94, 1.00) | float32 |

For FastSurfer and FONDUE, we used the Consortium for Reliability and Reproducibility (CoRR) dataset, a multi-centric, open resource aimed to evaluate test-retest reliability and reproducibility. We randomly selected 5 T1-weighted MRIs from 5 different subjects, one from each CoRR acquisition site, and accessed them through Datalad Halchenko et al. (2021). The selected images included a range of image dimensions, voxel resolutions and data types (Appendix Table 1). We processed all subjects' images using FreeSurfer's recon-all command with the following steps: `-motioncor -talairach -nuintensitycor -normalization -skullstrip -gcareg -canorm -careg`. These steps ensured that the images were motion-corrected, skull-stripped, intensity-normalized, and registered both linearly and non-linearly, preparing them as input for FastSurfer segmentation.

For the MNIST model, the CNN used in this experiment was custom-built while the dataset was downloaded from PyTorch's torchvision library.

For Xception, we evaluated the model on the first 1,000 samples of the ImageNet validation dataset Deng et al. (2009) available on Kaggle in order to further test our approach beyond neuroimaging and on convolution variations such as depth wise separable convolutions. Due to the current incompatibility of Adaptive Pooling and Linear layers with NaN propagation, NaNs were converted to zeros before the final two layers.

When applying NaN instrumentation to the models, we used Approach A within NaN Convolution for NaN substitution for the neuroimaging CNNs and Approach B for the MNIST CNN in order to achieve robust performance.

We processed the data for FastSurfer, FONDUE and Xception on the Anonymous cluster from Anonymous which include AMD Rome 7502, AMD Rome 7532, and AMD Milan 7413 CPUs with 48 to 64 physical cores, 249 GB to 4000 GB of RAM and Linux kernel 3.10. We executed the MNIST

CNN and the NaN Convolution time trials on the Anonymous cluster with 8 × compute nodes each with an Intel Xeon Gold 6130 CPU, 250 GB of RAM, and Linux kernel 4.18.0-240.1.1.el8_lustre.x86 64 as well as 3 Tesla T4 GPUs with 16GB of memory each. We used FreeSurfer v7.3.1, FONDUE v1.1, FastSurfer v2.1.1, PyTorch v2.4.0, and Singularity/Apptainer v1.2.

## C MODEL ARCHITECTURES

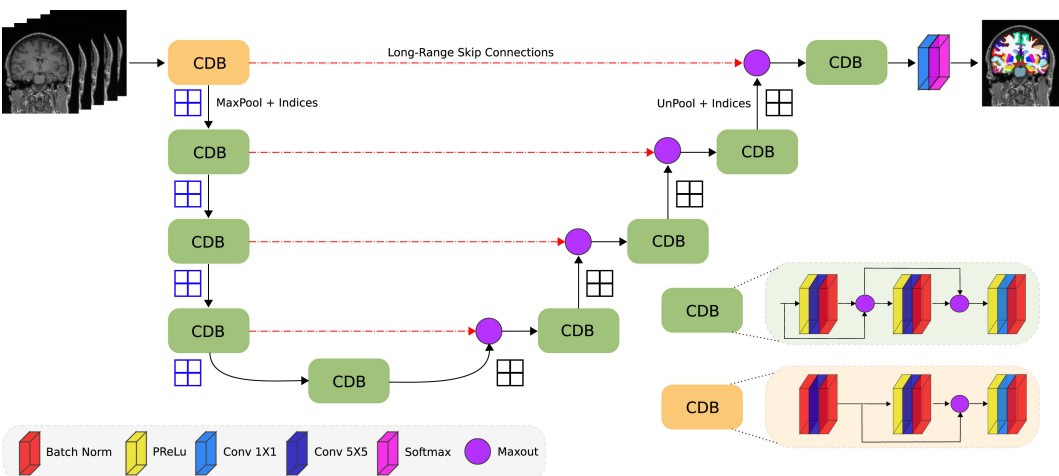

Figure 10: Illustration of FastSurfer's architecture. The CNN consists of four competitive dense blocks (CDB) in the encoder and decoder part, separated by a bottleneck layer. Figure reproduced from Henschel et al. (2020).

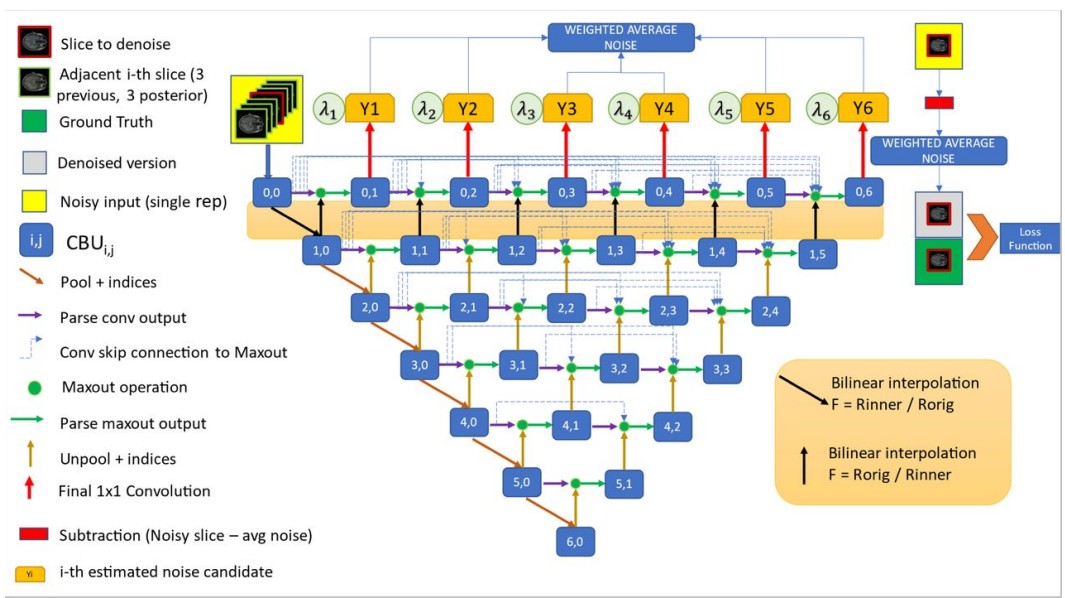

Figure 11: Illustration of FONDUE's architecture. The CNN consists of convolutional block units (CBU) in the nested encoder and decoder parts. Figure reproduced from Adame-Gonzalez et al. (2023).

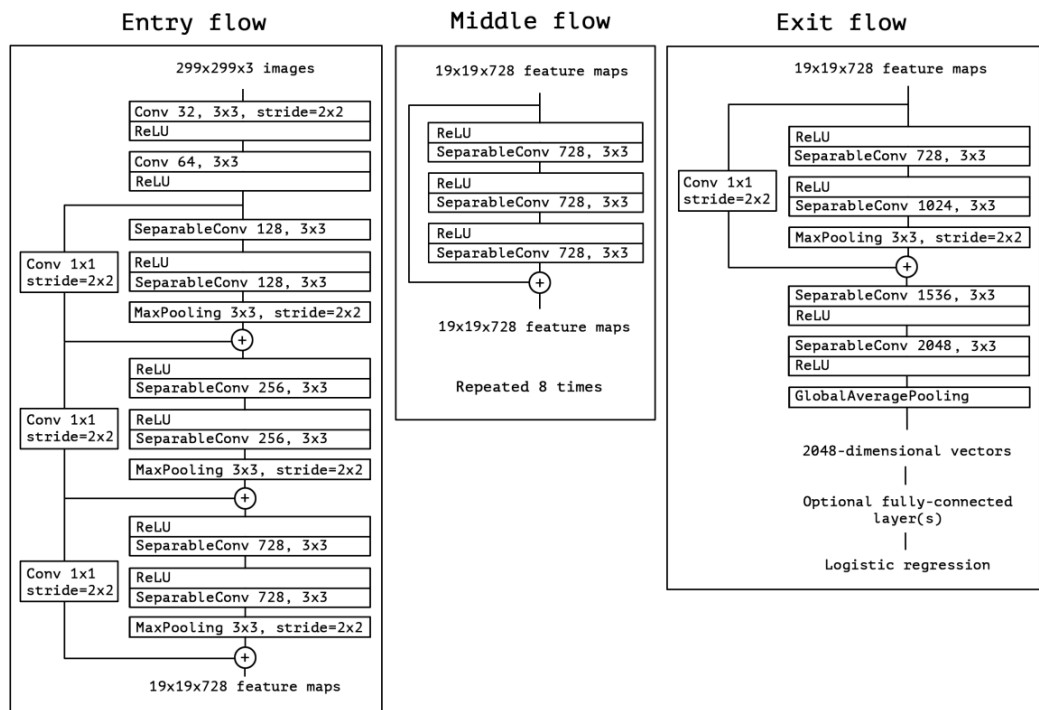

Figure 12: Illustration of Xception's architecture. The data first goes through the entry flow, then through the middle flow which is repeated eight times, and finally through the exit flow. Note that all Convolution and SeparableConvolution layers are followed by batch normalization (not included in the diagram). All SeparableConvolution layers use a depth multiplier of 1 (no depth expansion). Figure reproduced from Chollet (2017).

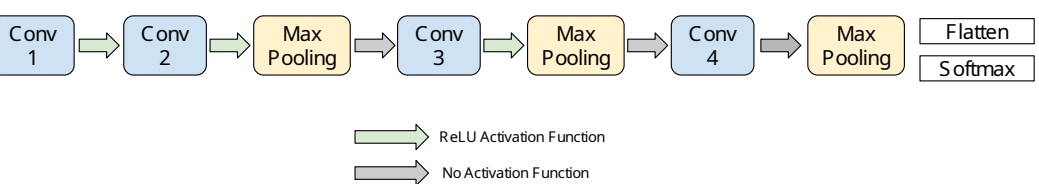

Figure 13: Illustration of the MNIST CNN's architecture.

## D   CEREBELLUM SEGMENTATION

A regional analysis (Appendix Figure 14) reveals that most of the degradation is concentrated in the cerebellum, a region known for its complex anatomy and segmentation challenges. Appendix Figure 15 highlights that FastSurfer, trained on FreeSurfer segmentations, inherits its limitations in this area. This suggests that segmentation errors in this region stem from the underlying dataset rather than Aggressive NaNs itself. Prior work has shown that the cerebellum is difficult to segment due to its intricate structure, proximity to other brain regions, high inter-subject variability, and often low contrast in neuroimaging data Morell-Ortega et al. (2024); Carass et al. (2018); Romero et al. (2017). Additionally, FreeSurfer segmentations, which FastSurfer was trained on, are known to struggle with these regions. Additional visualizations of cerebellum segmentation quality are provided in Appendix Figure 16.

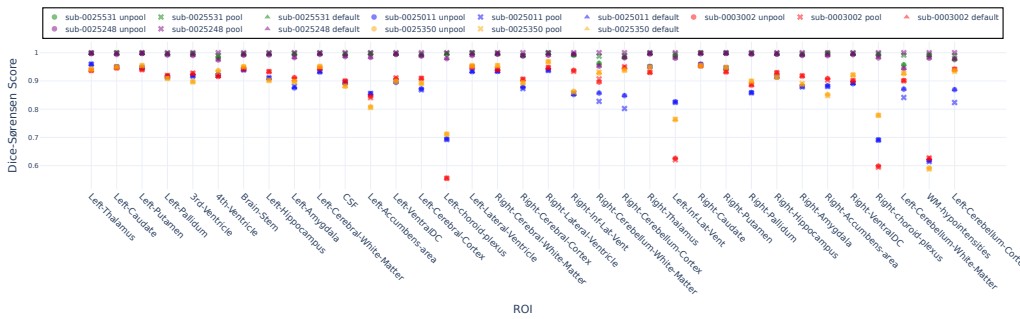

Figure 14: Dice-Sørensen Score Analysis of FastSurfer Across Default, Conservative NaNs (Unpool) and Aggressive NaNs (Pool) implementations for brain regions of interest (ROI). Threshold 0.5 was used for both NaN implementations here due to being the most stringent threshold common across models and methods. The legend maintains consistent colors for the same subjects across methods to enhance clarity.

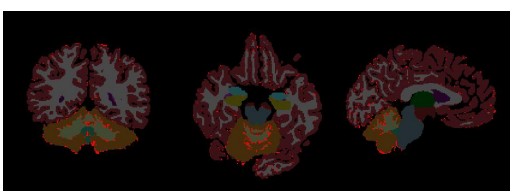
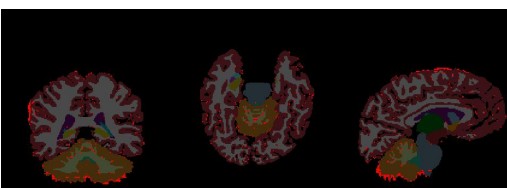

(a) Worst Performing Subject for Threshold 1; sub-0025248

(b) Worst Performing Subject for Threshold 0.5; sub-0025011

Figure 15: Comparison of segmentation outputs between Aggressive NaNs and default FastSurfer across different thresholds, displayed in coronal (left), axial (center), and sagittal (right) planes. The different brain regions are colored according to the FastSurfer colormap, except for the bright red voxels scattered throughout the brain which denote differences in segmentation outputs.

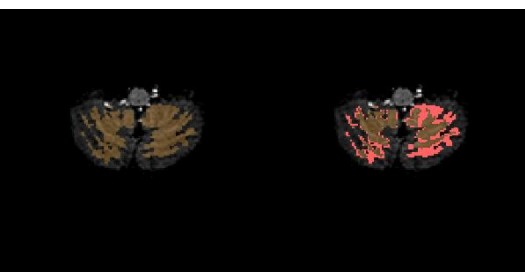

Figure 16: Comparison of FastSurfer's cerebellum segmentation with and without Aggressive NaNs. On the left is the default FastSurfer segmentation, while on the right, the overlay shows the differences between NaN-FastSurfer (threshold 1) and the default version. Both segmentations are superimposed on the anatomical MRI scan of the cerebellum for reference.

