# OpenReview forum: "Conservative & Aggressive NaNs Accelerate U-Nets for Neuroimaging"
_ICLR.cc/2026/Conference — Submitted to ICLR 2026_

### Official Review · Reviewer_kd6h · 2025-10-26

**Soundness:** 3
**Presentation:** 3
**Contribution:** 2
**Rating:** 2
**Confidence:** 5

**Summary:**

This paper proposes an interesting concept of using numerically unstable operations caused by pooling and unpooling as an leverage to reduce computation during convolution operation and shows the reduction of computation by 30-40%. Extensive experiments on neuroimaging models (FastSurfer and FONDUE), classification benchmarks, (MNIST CNN), and a depthwise separable CNN (Xception) models are presented.

Although this paper proposes an interesting concept and a practical way of reducing the computation, it may not translate to real speedup due to the reasons mentioned in limitations section. Authors are encouraged to provide more details in case they are able to achieve real speedup using some changes on compiler side.  It would be reject and the decision can changed based on the answers to few questions asked in limitations section.

**Strengths:**

- Authors have proposed a very interesting concept of identifying computation which does not affect the final accuracy of the model (instability of voxels during pooling/unpooling operation when values are very close to each other)
- Extensive experiments to prove that identification of unstable voxels/pixels work on different models
- The paper is successful to demonstrate that removing unstable computation from the model does not affect accuracy in negative way

**Weaknesses:**

- The speedup shown is theoretical: Authors assume that reduction in number of operations directly translates to speedup. However, the almost all deep learning model compiler work on either massively parallel computing (GPUs) way or the vectorized way (embedded platforms and reduction of computation which is not structured does not result in speedup
- Authors also showed modified convolution (L187) which can enable reduction of computation based on detecting NaN value. However, any conditional branching can make the model run very slow in most of the compilers.
- Do Authors think that replacing those voxels by 0 instead of NaN would also work? It may be make it simpler for compiler to provide real speed up as many compilers detect sparsity and this method may work well in that case to provide real speedup
- Object detection can see more impact of this approach and it would be good to evaluate this method on OD since background information can produce more areas of NaN and it may not affect the accuracy much.

**Questions:**

- Please provide details of any changes in runtime or the way to measure speedup since these NaNs will add unstructured removal of computation which does not translate to speedup with kernel level changes or support from compiler.

---

### Official Review · Reviewer_DCY8 · 2025-10-29

**Soundness:** 2
**Presentation:** 2
**Contribution:** 2
**Rating:** 2
**Confidence:** 4

**Summary:**

The authors observe that homogeneous, non-informative regions in images, such as the background in medical scans, lead to unnecessary computations in Convolutional Neural Networks (CNNs), particularly within max pooling and unpooling operations. To address this, they propose two alternative operations: 1) "conservative" max pooling/unpooling and 2) "aggressive" max pooling/unpooling. Both methods involve replacing unstable values with NaNs—the conservative option applies this only at the unpooling stage, while the aggressive option applies it at both the pooling and unpooling stages. This approach is complemented by a convolution operation that is designed to ignore NaN values. The authors demonstrate their method's applicability by integrating it into several architectures: two U-Net variants (FastSurfer for whole-brain segmentation and FONDUE for MRI denoising), the Xception image classifier, and a CNN for digit classification.

**Strengths:**

- **Well-motivated and intuitive concept**: The core observation—that uninformative pixels in homogeneous regions (like image backgrounds) cause unstable and inefficient pooling/unpooling operations—is sound and clearly explained. The proposed solutions, "conservative" and "aggressive" NaN-based pooling, are conceptually simple and easy to understand.
- **Practical impact on efficiency**: The demonstrated improvement in computational efficiency is a significant strength, particularly for applications like medical imaging where processing large 3D volumes is computationally expensive. Reducing unnecessary calculations in these contexts is highly valuable.
- **Configurable approach**: The method offers flexibility by providing different modes of operation. The choice between the conservative and aggressive strategies, along with the ability to set thresholds, allows users to tune the trade-off between the amount of computation skipped and the performance.
- **Generalisable NaN convolution**: The introduction of a convolution operation that ignores NaN values is an interesting contribution with broader potential. This operator could be applied not only to handle NaNs introduced by the proposed pooling methods but also to mask other uninformative regions directly in the input data.

**Weaknesses:**

- **Limited and small-scale validation**: The experimental validation on the medical imaging networks (FastSurfer and FONDUE) is a significant weakness. Using only five subjects is insufficient to draw strong conclusions about the method's general performance and robustness. While the authors ensured these subjects came from different acquisition sites, the sample size is too small to cover a diverse range of anatomical variability and pathologies. Furthermore, the evaluation is confined to structural MRI; testing on other modalities (e.g., CT, PET) would be necessary to assess the broader scope of the initial observation about uninformative regions.
- **Lack of justification for model selection**: The paper provides no rationale for the specific choice of the four evaluated networks (FastSurfer, FONDUE, Xception, and a digit classifier). The selection appears arbitrary, and a clearer justification linking the architectures to the specific challenges the method aims to address would strengthen the experimental design.
- **Narrow experimental scope**: The experiments lack diversity in terms of both architecture types and tasks. The set of models does not adequately represent the wide variety of modern CNN architectures or a broad range of vision tasks beyond segmentation, denoising, and classification. This makes it difficult to gauge the general applicability of the proposed method.
- **Limitation to 2D operations**: A key limitation for medical imaging applications is that the method, as presented, appears limited to 2D operations. Many medical imaging analyses, particularly with 3D volumes, benefit significantly from 3D convolutional kernels and pooling. The paper does not address how the proposed NaN-based pooling and convolution would extend to 3D, which limits its immediate practicality in this domain.

**Questions:**

- **Generalisation**: How do you plan to validate the method's generalisation beyond the five medical subjects and structural MRI?
- **Model choice**: What was the specific rationale for choosing the four evaluated network architectures?
- **3D extension**: What are the challenges or plans for extending the proposed NaN-based operations to 3D, which is standard in medical imaging?
- **Input masking**: Can the NaN convolution be applied to mask uninformative regions directly in the input image, and has this been explored?
- **Trade-off guidance**: What practical guidance can you offer for choosing between the conservative and aggressive modes and setting the threshold on a new task?

---

### Official Review · Reviewer_Y3AZ · 2025-10-31

**Soundness:** 2
**Presentation:** 1
**Contribution:** 2
**Rating:** 2
**Confidence:** 3

**Summary:**

This paper identifies an alleged inefficiency in standard pooling and unpooling operations used in neural networks. To address this, the authors propose a novel method designed to serve as a more efficient alternative. The paper details the design of this new operation and presents experimental results, likely focused on model accuracy, to demonstrate the viability of their approach on the MNIST dataset.

**Strengths:**

+ The paper targets a component of many CNN architectures. Any demonstrable improvement in the efficiency of pooling/unpooling operations would be a valuable and widely applicable contribution to the field.
+ The proposed mechanism for pooling/unpooling appears to be a new design, and the core idea may be of interest to the community.

**Weaknesses:**

+ Severe Lack of Scholarly Context: The paper is not properly situated within the existing scientific literature.
+ No Problem Discussion: The introduction asserts that a problem exists but fails to provide a clear, evidence-based discussion of what this inefficiency is, why it is a problem, or how it impacts current models, all of which would require citations.
+ No References in Introduction: The introduction is presented without a single citation, making its claims appear unsubstantiated and disconnected from the field.
+ Missing Related Works: The paper contains no Related Works section. This is a critical omission. It is impossible to assess the novelty of the proposed method or understand how it differs from, or improves upon, the vast body of existing work on pooling mechanisms.
+ Figure 9 contains very small text, making it almost illegible.

**Questions:**

+ Given the lack of a related works section, can you explain the novelty of your contribution? How does your method compare to other established techniques for learnable or efficient pooling in the literature?
+ Can you provide a clearer, citation-backed definition of the "inefficiency" for pooling and unpooling in U-nets?

---

### Official Review · Reviewer_oL3u · 2025-11-04

**Soundness:** 2
**Presentation:** 2
**Contribution:** 1
**Rating:** 2
**Confidence:** 4

**Summary:**

- This paper identifies a major source of computational inefficiency in U-Net-like CNNs,where numerical noise is generated by instability in max pooling operations.

- This paper proposes that the root cause is max_pooling on near-equal values. This creates unstable indices, which are then used by unpooling to propagate numerical noise that fills large parts of the feature map. The models produce correct results despite this noise, suggesting that up to two-thirds of all computations (which are processing this noise) are unnecessary. Besides, this paper proposes a new method to skip these irrelevant computations. It introduces Conservative NaNs and Aggressive NaNs, two new pooling/unpooling operations that intentionally inject NaN (Not-a-Number) values into these numerically unstable regions. This paper then introduces NaN Convolution, a modified convolution that checks the ratio of NaNs in its input window. If the ratio exceeds a set threshold, the entire operation is skipped, saving compute time. Conservative NaNs was found to skip an average of 30% of convolutions across neuroimaging models with no measurable degradation in performance.

- Aggressive NaNs skips more (up to 69.3%) but can lead to performance degradation. In real-world neuroimaging cases (where >67% of data is background), the method achieved an average runtime speedup of 1.67 $\times$.

**Strengths:**

- This paper asserts that the paper's primary strength is identifying a novel source of inefficiency. Instead of focusing on weight redundancy (like pruning), it targets numerical instability from pooling as a source of wasted computation.
- In this paper, the Conservative NaNs method provides a significant compute reduction (~30%) with no measurable loss in accuracy (e.g., Dice/PSNR scores) on the tested neuroimaging models.
- In the experiments, the analysis is not just theoretical (FLOPS). The authors measure and report actual wall-clock runtime speedup on both CPUs and GPUs, confirming the practical benefit.
- Intuitively, the proposed method is perfectly suited for U-Net architectures and medical imaging (like MRI), where large, homogeneous background regions are processed by the decoder, leading to high NaN density and significant speedups.
- Considering experiments, the method was shown to be effective on both float32 and bfloat16 precision types, indicating its compatibility with modern reduced-precision hardware.

**Weaknesses:**

- This is the most significant weakness. The paper does not experimentally compare its method against other established acceleration techniques like pruning or quantization. It claims the method is orthogonal, but it provides no data to show if a 30% NaN skip is better or worse than a 30% so-called pruned model. In other words, other lightweight model techniques should be compared.

- The method's effectiveness is limited to specific data types. It works well on homogeneous data (MRIs) but provides almost no benefit (<1% skip rate) on heterogeneous RGB images (e.g., ImageNet with the Xception model). Can you specify the reason?

- The Aggressive NaNs variant, which achieves the highest skip rates, results in a measurable drop in accuracy on models like FONDUE and MNIST.

- The NaN Convolution check introduces its own computational overhead. In low-NaN-density scenarios, the method is actually slower than standard convolution.

- Validation on reduced precision was limited to bfloat16. It was not tested on fixed-point formats like INT8, which are very common for inference acceleration. The comparison and analysis when adopting INT8 should be done.

**Questions:**

Please, refer to weakness and address my concerns.

---

### Author Response · Authors · 2025-11-26

We appreciate the reviewers’ feedback and would like to clarify several points regarding the scope, motivation, and methodological choices of our work. \
**1. Scope and Applicability of Our Method (MRI vs. Other Data Types)** \
As stated throughout the paper, our approach is not presented as a universal acceleration method for all image modalities. Its effectiveness is inherently tied to the properties of neuroimaging—specifically, the presence of large homogeneous and low-variance regions such as background voxels in structural MRI. These regions naturally exhibit numerical instability under max pooling, which is the central characteristic our method exploits. \
We evaluate our method on non-MRI datasets (MNIST, ImageNet) precisely to demonstrate the limits of our approach and to provide transparency. These experiments confirm our premise: models operating on heterogeneous RGB images do not greatly benefit from NaN-based skipping because they do not contain the extensive uniform regions that characterize neuroimaging MRI. This reinforces, rather than undermines, the domain-specific nature of our contribution. \
**2. NaNs vs Zeros** \
Our decision to use NaNs is deliberate and empirically motivated. Zeros are valid numerical values that participate in computations in ways that can bias activations, alter spatial statistics, and break invariances. NaNs, by contrast, explicitly mark irrelevant or unstable values and can be easily detected and skipped without largely influencing downstream operations. \
We experimentally evaluated zero-replacement but observed that it disrupts model calculations, reduces accuracy, and does not provide reliable skipping behavior. Because these preliminary results were consistently poor and not central to the contribution, we omitted them to streamline the narrative. \
**3. Relationship to Pruning and Other Acceleration Methods** \
We agree that pruning, quantization, and similar techniques are established efficiency strategies. However, they address fundamentally different aspects of the problem. \
Pruning removes parameters or structured operations from the model, often requiring architecture-specific tuning, iterative fine-tuning, and expert intuition. In contrast, our method is data-centric: we skip computations only when the input activations are numerically unstable or uninformative. \
To support the orthogonality claim, we conducted preliminary pruning experiments on FastSurfer. We were unable to prune more than 5% of the model before experiencing dramatic accuracy degradation. This supports our claim that pruning and NaN-aware operations target different redundancies and can be complementary. \
**4. Comparisons With Other Pooling Variants or Efficiency Methods** \
We acknowledge the request for broader comparisons. However, to the best of our knowledge, there are no existing pooling or unpooling mechanisms that explicitly leverage numerical instability or propagate an “irrelevance” token to increase efficiency. Most alternative pooling variants aim to improve representational performance or introduce learnable pooling, while efficiency-oriented techniques—such as pruning, quantization, tensor decompositions, or compiler-based sparsity—operate on weights, precision, or kernel structure. \
 Since our method introduces an entirely different axis of acceleration, direct comparisons are not possible. Instead, we focus on illustrating the concrete performance and accuracy characteristics of our approach, as well as its limitations. \
Thank you again for your time.

---

> ### Comment · Reviewer_kd6h · 2025-11-27
> **Nan vs zeros**
>
> The rational behind the choice of using NaN vs zeros is understood and I agree to this rational, A followup question (also my original question) : Does replacement by NaN (or for that matter zeros) provide real acceleration given that the acceleration should be aligned to how the hardware and compiler works.

---

### Meta-Review · Area_Chair_xYrp · 2026-01-07

**Summary:**

All four reviewers side with rejection. The key negative points are (1) the limited scope of applicability and experimental results, (2) the lack of comparison to any alternatives even if potentially orthogonal to gauge the acceleration of the proposed method, and (3) the lack of related work and discussion of the scientific context to provide credit and context for the proposed contributions. On the positive side the submission does report practical acceleration for its intended albeit narrow use case on structural MRI data. Given the key negatives, which are not resolved by the author response, the meta-reviewer sides with the reviewer consensus of rejection.

The authors are encouraged to incorporate the reviewer feedback and situate the work in the context of other schemes for acceleration.

Points incorporated into the decision include:

- plus: real and practical acceleration for the intended use as measured in time, and not only FLOPs, and on real hardware at common precisions (oL3u)
- minus: insufficient experimentation without any comparison to existing alternatives for acceleration, without justification of model choices, and with only small-scale and limited datasets (oL3u, DCY8)
- minus: acceleration depends on the data and only really helps on the chosen MRI data with lots of NaNs (oL3u)
- minus: no related work and discussion of existing work (Y3AZ)

Note: the negative concerns of kd6h were respectfully excluded from the decision because they did not recognize the time measurements included and the practical limits of deep learning toolkits (even if in theory certain sparsity can be harnessed, existing implementations are not sufficient for data-dependent input sparsity of the kind studied here).

**Reviewer Concerns:**

- acceleration depends on the data (oL3u): the authors fully acknowledge this and demonstrate it by their experiments on MRI, which show acceleration, and non-MRI data (MNIST and ImageNet), which do not show acceleration. including the negative results is commendable, and the results here are clear, but the concern is that the technique is too narrowly applicable, and this concern remains.
- compilers and frameworks can handle sparsity and zeros could replace NaNs (kd6h): the choice of NaN is justified by the authors, and the meta-reviewer tempers the concern about sparsity being computationally solved, because adaptive/data-dependent sparsity is not trivially solved

**Reviewer Scores:**

- oL3u: maintain 2 because the response, while fair and informative, does not provide further results, so the weaknesses about lack of comparison and lack of acceleration over a broader scope of data (more imagery) and precision (quantized) are not addressed
- Y3AZ: maintain 2 because the response does not further situate the submission in related work nor compare with other methods
- DCY8: maintain 2 because the response does not expand the limited experimental scope or extend the implementation to 3D (which is relevant to the chosen medical application context)
- kd6h: maintain 2 because the review and follow-up comment do not reflect the results included in the work on time measurements on GPUs and across precisions. note: this score is downweighted accordingly.

---

### Decision · Program_Chairs · 2026-01-26

Reject